# Design and Implementation of Sigma-Delta ADC Filter

**Renzhuo Wan [1], Yuandong Li [1], Chengde Tian [1], Fan Yang [2], Wendi Deng [1], Siyu Tang [2], Jun Wang [1] and Wei Zhang [1,\*]**

[1]  School of Electronic and Electrical Engineering, Wuhan Textile University, Wuhan 430200, China
[2]  School of Mathematical and Physical Sciences, Wuhan Textile University, Wuhan 430200, China
\*  Correspondence: wzhang@wtu.edu.cn

**Abstract:** This paper presents a digital decimation filter based on a third-order four-bit Sigma-Delta modulator. The digital decimation filter is an important part of the Sigma-Delta ADC and is designed to make the Sigma-Delta ADC (Analog-to-Digital Converter) meets the requirements of Signal-to-Noise Ratio (SNR) not less than 120 dB and Equivalent Number of Bits (ENOB) not less than 20 bits. It adopts a three-stages cascaded structure including a Cascaded Integrator Comb (CIC) decimation filter, a Finite Impulse Response (FIR) compensation filter, and a half-band (HB) filter. This structure effectively reduces about 13% multiplier cells and memory cells. The coefficient symmetry technique and CSD (Canonic Signed Digit) coding technique are used to optimize the parameters of the filter, which further reduces the computational complexity. After optimization, the circuit area is reduced by about 15%, and the logic resources are decreased by about 23%. The Verilog hardware description language is used to describe the behavior of the digital decimation filter, and the simulation is carried out based on the VCS (Verilog Compile Simulator) platform. At the same time, the prototype verification is implemented on the Xilinx Artix-7 series FPGA, and the ADC achieves 113 dB SNR and 18.5 bits ENOB. Finally, the Sigma-Delta ADC is fabricated on SMIC 0.18 μm CMOS process with the layout area of 714.8 μm × 628.4 μm and the power consumption of 11.2 mW. The more tests for the fabricated prototypes will be performed in the future to verify that the Sigma-Delta ADC complies with the design specifications.

**Keywords:** sigma-delta ADC; digital decimation filter; CIC decimation filter; compensation filter; half-band filter

## 1. Introduction

As digital information technology develops by leaps and bounds, the precision of signal processing in medical, manufacturing, aerospace and other fields is getting higher and higher [1], and the conversion precision also becomes an important parameter in digital signal processing [2]. Sigma-Delta ADC has the characteristics of low power consumption, high accuracy and high linearity. It has been widely used in sensor monitoring, precision measurement, modern voice band and audio applications [3–8].

In order to obtain higher conversion accuracy, ADC will also develop towards higher performance and higher resolution. Due to the limitations of chip technology, the conversion accuracy of integral ADC, successive approximation ADC, parallel comparison ADC, and capacitor array successive comparison, ADC has reached the bottleneck and cannot be further improved [9]. The Sigma-Delta ADC can conquer the limitations of chip technology due to over-sampling technology and noise shaping technology, which makes ADC accuracy and performance more improved [10].

The Sigma-Delta ADC is composed of a Sigma-Delta modulator and a digital decimation filter [11]. Power consumption, area, and conversion accuracy are very important technical specifications of the Sigma-Delta ADC. The accuracy is mainly determined by the Sigma-Delta modulator, and the area and power consumption are mainly determined by the digital decimation filter [12]. With the development of Sigma-Delta ADCs, the SNR

of Sigma-Delta modulator has reached more than 120 dB [13], which will put forward higher performance requirements for the design of digital decimation filter [14]. When the structure of the digital decimation filter is determined, it is necessary to consider the effect of stop-band suppression and the requirements of the design specifications. Because the high-speed and low resolution digital signals generated by the Sigma-Delta modulator contain a lot of quantization noise [15], it is difficult for a single-stage filter to suppress the stopband well, and the passband is usually about 10 dB attenuation [16]. Therefore, many designs adopt multistage filter structure, and CIC decimation filter and anti-aliasing filter are commonly used for de-sampling and filtering [17]. Similarly, there are also designs to replace FIR filter with Infinite Impulse Response (IIR) digital filter. Compared with FIR filter, IIR filter is simpler in calculation and smaller in area. However, the IIR filter has a nonlinear phase, so the additional phase compensator is required [18]. The design adopts a three-stages cascaded filter structure. The first stage is a CIC decimation filter, the second stage is composed of FIR compensation filter, and the third stage consists of half-band filter. With the structure and coefficient optimization, a digital decimation filter is designed to meet the requirements.

This paper presents design details of a fully synthesized digital decimation filter for the Sigma-Delta ADC. In Section 2, Sigma-Delta ADC architecture is discussed. RTL level implementation of the digital decimation filter is reviewed in Section 3. Finally, FPGA prototype verification, layout of the digital decimation filter, and conclusions are presented in Sections 4–6.

## 2. Sigma-Delta ADC Architecture

The block diagram of Sigma-Delta ADC which involves a Sigma-Delta modulator and a digital decimation filter is represented in Figure 1. The Sigma-Delta modulator consists of a third-order modulator, a loop filter, and a DAC. It modulates analog signals into high-speed, low resolution oversampling signals, pushes quantization noise to high frequency, and thus achieves high SNR and ENOB. The digital decimation filter is made up of a digital filter and a decimator. The digital filter filters and anti-aliases the oversampled signal containing quantization noise. The decimator decimates the output signal of the Sigma-Delta modulator to reduce the sampling frequency of the signal and make it output at the Nyquist rate.

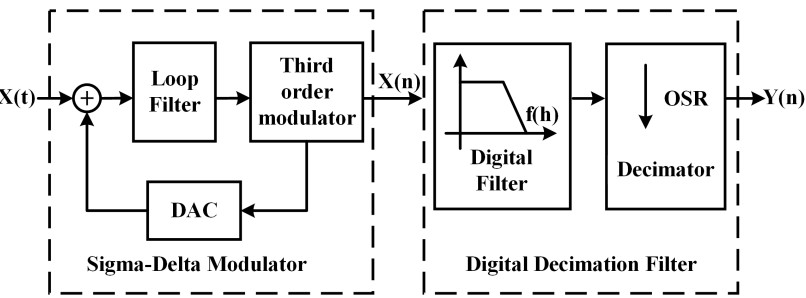

**Figure 1.** Block diagram of the Sigma-Delta ADC.

### 2.1. Sigma-Delta Modulator

Sigma-Delta modulator is the core module of ADC to realize oversampling and noise shaping. Analog signals are discretely output into high-speed, low resolution oversampled signals through Sigma-Delta modulator, and most of the noise power is pushed to high frequency to achieve high SNR. Sigma-Delta modulator can be divided into first-order, second-order, high-order topology, and MASH structure. Compared with the topology above the third order, the first-order and second-order modulator structures have the advantages of less influence of incomplete matching and simple circuit design but cannot meet the requirement that the SNR is not less than 120 dB. Compared with a single-bit quantizer, a multi-bit quantizer has lower quantization noise and higher loop stability, but requires higher area cost and power consumption. Combined with the comprehensive

consideration of design complexity, power consumption, area and other indicators, this design was selected a third-order four-bit Sigma-Delta modulator to achieve a SNR of about 120 dB. The block diagram of the third-order Sigma-Delta modulator is depicted in Figure 2.

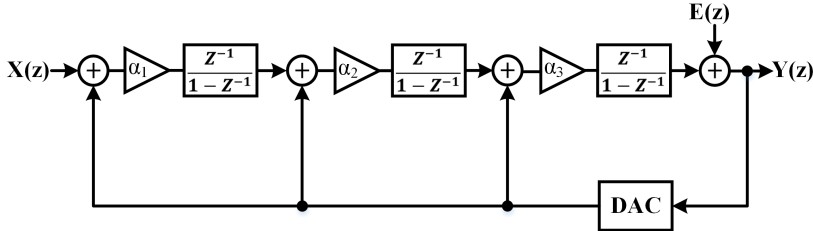

**Figure 2.** Block diagram of the third-order Sigma-Delta modulator.

For the third-order four-bit Sigma-Delta modulator, the system transfer function $Y(z)$ is derived from the superposition theorem, as given in Formula (1):

$$Y(z) = STF(z) * X(z) + NTF(z) * E(z) \tag{1}$$

$$STF(z) = \frac{(a_1 a_2 a_3) z^{-3}}{(a_1 a_2 a_3 - a_2 a_3 + a3 - 1)z^{-3} + (a_2 a_3 + 3)z^{-2} + (a_3 - 3)z^{-1} + 1} \tag{2}$$

$$NTF(z) = \frac{(1 - z^{-1})^3}{(a_1 a_2 a_3 - a_2 a_3 + a3 - 1)z^{-3} + (a_2 a_3 + 3)z^{-2} + (a_3 - 3)z^{-1} + 1} \tag{3}$$

In Formula (1), $X(z)$ represents the input signal, $E(z)$ denotes the quantization noise, $STF(z)$ is the signal transfer function (Formula (2)), $NTF(z)$ is the noise transfer function (Formula (3)), and $a_1, a_2$, and $a_3$ are constant.

The noise of the Sigma-Delta modulator mainly comes from the quantization noise. Assuming that the quantization step is $\Delta$ and the quantization noise is uniformly distributed on $(-\Delta/2, \Delta/2)$, the average power of the quantization noise is its variance. The power of quantization noise $P_{noise}$ is given in Formula (4):

$$P_{noise} = \frac{\Delta^2}{12} * \frac{\pi^{2L}}{(2L + 1) * OSR^{2L+1}} \tag{4}$$

Assuming that the input signal is a sine signal with amplitude $V_s$, the input signal power is given in Formula (5):

$$P_{signal} = (V_s/\sqrt{2})^2 = V_s^2/2 \tag{5}$$

For the third-order four-bit Sigma-Delta modulator, the bit $N$ of the output signal is 4, and the quantization step is $\Delta$. The relationship between $\Delta$ and $N$ is given in Formula (6):

$$\Delta = 2V_s/2^N = V_s/2^{N-1} \tag{6}$$

Obtain the signal noise of the output signal of the Sigma-Delta modulator as given in Formula (7):

$$SNR = 10 \lg \left(\frac{P_{signal}}{P_{noise}}\right) = 6.02N + 1.76 + 10 \lg \left(\frac{7}{\pi^6} * OSR^7\right) \tag{7}$$

The *OSR* is the abbreviation of oversampling rate that is calculated in the following Formula (8):

$$OSR = f_{sampling}/f_{Nyquist} = f_s/2f_b = 128 \tag{8}$$

### 2.2. Digital Decimation Filter

Digital decimation filter is an important part in the Sigma-Delta ADC, and its structure has a significant impact on design complexity, power consumption, and area. On the basis of the third-order four-bit Sigma-Delta modulator, according to the principles of cost requirements, market demand, design requirements, simplicity, and easy realization, this paper compares a variety of digital filters, and finally chooses the cascade filter structure as shown in Figure 3. The first stage is composed of a CIC decimation filter, the second stage is a FIR compensation filter, and the third stage consists of a half-band filter.

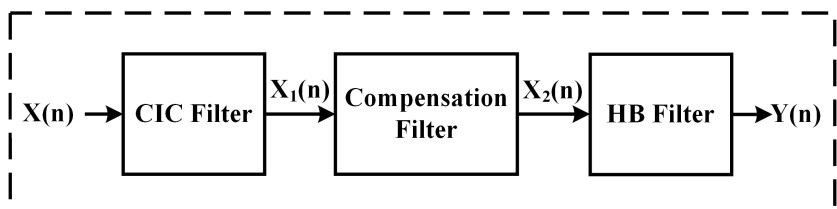

**Figure 3.** Block diagram of the digital decimation filter.

### 2.2.1. CIC Decimation Filter

CIC decimation filter is an important part of the digital decimation filter, which is mainly used for signal recovery, anti-aliasing, and de-sampling. Due to the CIC decimation filter having the characteristics of all-tap coefficients being 1, high decimation multiple and simple structure unit, it simplifies RTL level design and saves circuit hardware resources and area.

Based on the third-order four-bit Sigma-Delta modulator, the digital signal with the sampling frequency of 6.144 MHz will be output according to the design requirements, and the signal with the sampling frequency of 48 kHz will be output through the digital decimation filter, so the down-sampling rate of the digital decimation filter is 128.

According to the design requirements of 128 times down-sampling, and by comparing various structural characteristics, the CIC decimation filter adopts a recursive structure to conduct 32 times down-sampling on the output signal of the Sigma-Delta modulator. For the order of CIC decimation filter, it is necessary to refer to the order of the modulator structure. The relationship between the order of CIC decimation filter $N$ and the order of modulator $L$ is illustrated in Formula (9). The order of Sigma-Delta modulator is 3, so the minimum value of $N$ is 4. Therefore, the structure of CIC decimation filter is displayed in Figure 4.

$$N \geq L + 1 \tag{9}$$

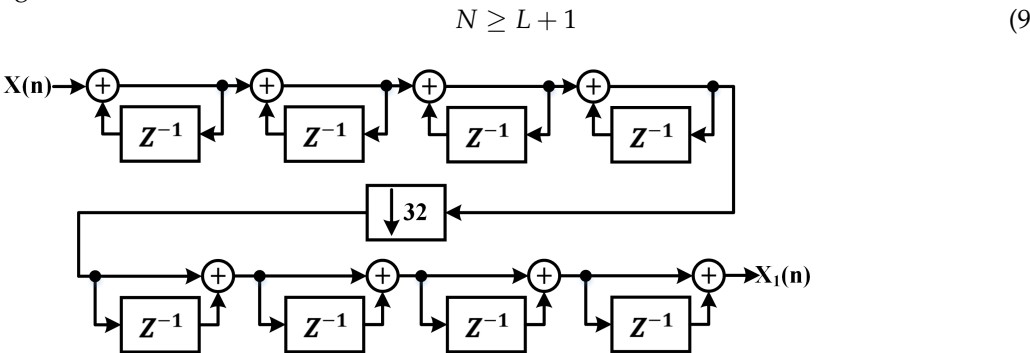

**Figure 4.** Structure of the CIC decimation filter.

According to Figure 4, the z-domain response of CIC decimation filter is calculated using the following Formula (10):

$$H(z) = \left( \frac{1 - z^{-32}}{1 - z^{-1}} \right)^4 \tag{10}$$

### 2.2.2. FIR Compensation Filter

The CIC decimation filter has a very good filtering effect with minimized delay units and adder units, while reducing the signal sampling frequency, effectively reducing hardware resources and power consumption. However, it also causes the attenuation of the signal in the passband (called passband roll off), and using Formula (8) calculates the amplitude frequency characteristics of the CIC decimation filter. Its passband ripple has about 1 dB attenuation, which does not meet the design requirement that the passband attenuation should be less than 0.01 dB.

With the increase of processing signal frequency and decimation multiple, the passband attenuation becomes more and more important. In order to offset the passband attenuation of CIC decimation filter, a compensation filter whose frequency response characteristics are reciprocal to CIC decimation filter's frequency response characteristics is required to realize the correction of frequency response.

The product of the amplitude–frequency response of the compensation filter and the CIC decimation filter is 1, so the amplitude–frequency response of the compensation filter can be obtained according to the amplitude–frequency response of the CIC decimation filter, as given in Formula (11):

$$\left| H(e^{jw}) \right| = \left| \frac{32 \sin (w/2)}{\sin (32 * w/2)} \right|^4 \tag{11}$$

Let us take $w = 2\pi f/32$ and substitute it into Formula (10), so the amplitude–frequency response of FIR compensation filter can be expressed by inverse sine function, as given in Formula (12):

$$|H(f)| = \left| \frac{32 \sin \pi f/32}{\sin (\pi f)} \right|^4 \approx \left| \frac{\pi f}{\sin (\pi f)} \right|^4 = \left| \sin c^{-1}(f) \right|^4 \tag{12}$$

The compensation filter has three structures: direct type, transposed type, and polyphase type. The coefficient coding technology also has three coding methods: Gray code, CSD code [19], and One-hot code [20]. The structure is compared in terms of implementation difficulty, number of multipliers, logic resources, layout area, and other indicators. Finally, the polyphase structure compensation filter is chosen, and the coefficient symmetry and CSD coding technology are used to optimize it. The structure of the multiphase structure compensation filter is shown in Figure 5.

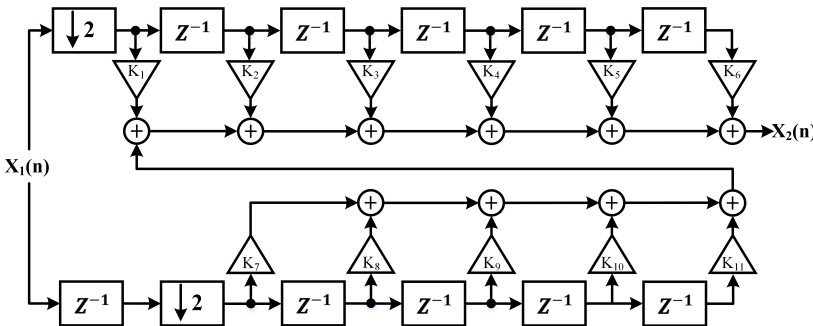

**Figure 5.** Structure of the multiphase compensation filter.

### 2.2.3. Half-Band Filter

The half-band filter is a special FIR filter. Compared with the traditional FIR filters, the half-band filter has the characteristics of flat passband, easy to control the bandwidth of the transition band, zero odd coefficients, and good anti-aliasing effect. These characteristics make the design of the half-band filter simple and convenient, and also reduce the hardware resources, power consumption, and area. Therefore, the last stage filter of decimation filter utilizes the half-band filter.

In order to simplify the design process, the half-band filter was designed using Matlab FDAtool, which uses the equal ripple design method. In the FDAtool, the filter order is 26, the default value of the density factor is 20, the sampling frequency is 48 kHz, and the passband frequency ranges from 0 to 5 kHz. In order to meet the design requirement that stopband attenuation reaches 120 dB, the passband ripple coefficient (the difference between the maximum amplitude and the minimum amplitude of the passband in the frequency response of the filter) is set to 0.00002 based on the simulation.

Compared with the direct type, the transposed type, and the polyphase type, combined with the characteristics of the half-band filter, the half-band filter uses the polyphase structure. Base on the FDAtool, coefficient symmetry and CSD coding technology are used to optimize the half-band filter. The center position unit of the half-band filter is single extracted to form an independent branch. Each delay unit of the rest is multiplied by the coefficient factor, and then added to obtain its specific structure as shown in Figure 6.

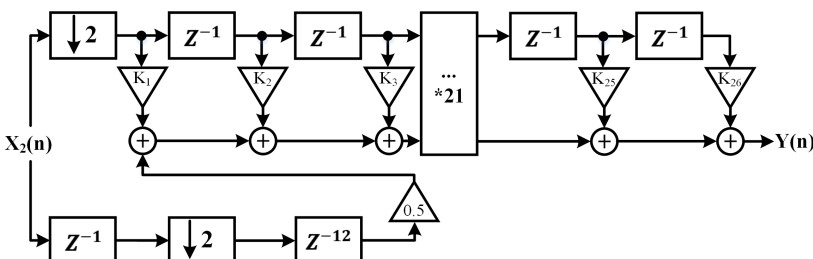

**Figure 6.** Structure of the multiphase half-band filter.

Since the coefficient of the central location unit is 0.5, and the other coefficients are less than 0.5, an independent branch is formed and added to the signal path. This model reduces the multiplier unit and memory unit by about 13% in structure, thus reducing the calculation amount of the operation unit and the power consumption.

## 3. RTL Level Implementation of the Digital Decimation Filter

The structure of the three-stage filter model was simulated and optimized using FDAtool. Finally, the CIC decimation filter adopted a 4-order recursive structure, the FIR compensation filter was designed as a 10-order polyphase structure, and the half-band filter was designed as a 26-order polyphase structure. According to the logic function of the filter, it can be described in RTL behavior level using Verilog, and its performance is simulated based on VCS+Verdi platform. In the entire simulation platform, the input signal is the output of the four-bit Sigma-Delta modulator, which is transmitted to the top module of the digital decimation filter through the top level input interface of the Testbench, and then output through the CIC decimation filter, the compensation filter, and the half-band filter. The output signal is saved to the data file in the standard IIS data format. Finally, the FDAtool decodes the output signal time domain waveform as shown in Figure 7a, performs FFT on it, and obtains the spectrum as shown in Figure 7b.

It can be seen from Figure 7a that the frequency of the output signal is 1 KHz, and the sampling frequency is 48 KHz (48 data points in a period), which meets the design requirements of the final output signal sampling frequency of 48 KHz. The four-bit modulated signal output by the Sigma-Delta modulator is restored to a sinusoidal signal, and meets the requirements of the final output signal sampling frequency of 48 KHz, which proves the correctness of the functional design. It can be seen from Figure 7b that the signal attenuation at 1 kHz is about 1 dB, and there is no obvious high-order harmonic distortion. The SNR of output signal is about 121 dB, and the ENOB is approximately 19.8 bits, which almost meet the design requirements.

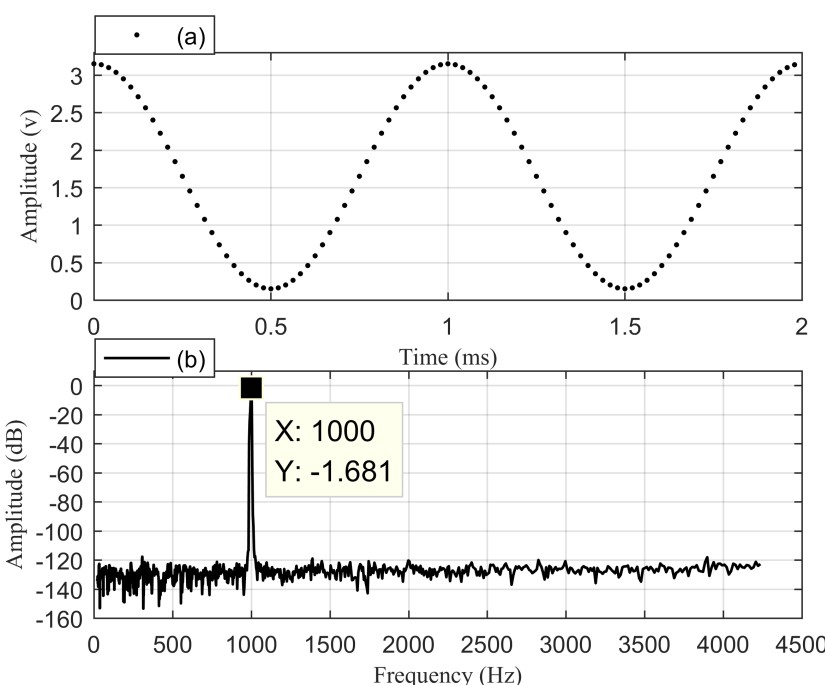

**Figure 7.** Simulated output signal waveform (**a**), output signal spectrum (**b**) of the digital decimation filter.

## 4. FPGA Prototype Verification

We introduced the model simulation of digital decimation filter using the FDAtool, and the functional simulation of digital decimation filter RTL code based on the VCS+Verdi verification platform. The simulation results show that the functions and performance of the decimation filter meet the design requirements, but the effects of physical routing, parasitic parameters, and PVT (Process Voltage Temperature) are not considered. Before the digital backend design of digital decimation filter, it is necessary to carry out prototype verification in FPGA.

This verification is carried out on the Xilinx Artix-7 series FPGA, and the photograph of the verification platform is shown in Figure 8. The RTL code is integrated and implemented in the Vivado IDE, where the logical resource utilization is 2% (2716/133,800), the IO port utilization is 3.5% (10/285), and the BRAM utilization is 1% (2/365). The generated bit file is burned into the FPGA. The input signal of the FPGA is the output signal of the Sigma-Delta modulator chip, which is internally de-sampled and filtered, and the output signal is sampled and then saved in the standard IIS data format. After the output signal is decoded and FFT, the spectrum of the output signal is obtained as displayed in Figure 9.

It can be seen from Figure 9 that the output signal attenuation at 1 kHz is about 3 dB, which is caused by the superposition of Sigma-Delta modulation chip attenuation and filter passband attenuation. In addition, harmonic distortion exists at 2 kHz, 3 kHz, and 4 kHz. This phenomenon is caused by many factors, such as the noise induced by the PCB, the interference of nonlinear components, and the low power supply rejection of the Sigma-Delta modulation chip under the real verification environment. Therefore, the final calculated SNR of the output signal is nearly 113 dB, and the ENOB is about 18.5 bits. The performance comparison with similar works is listed in Table 1.

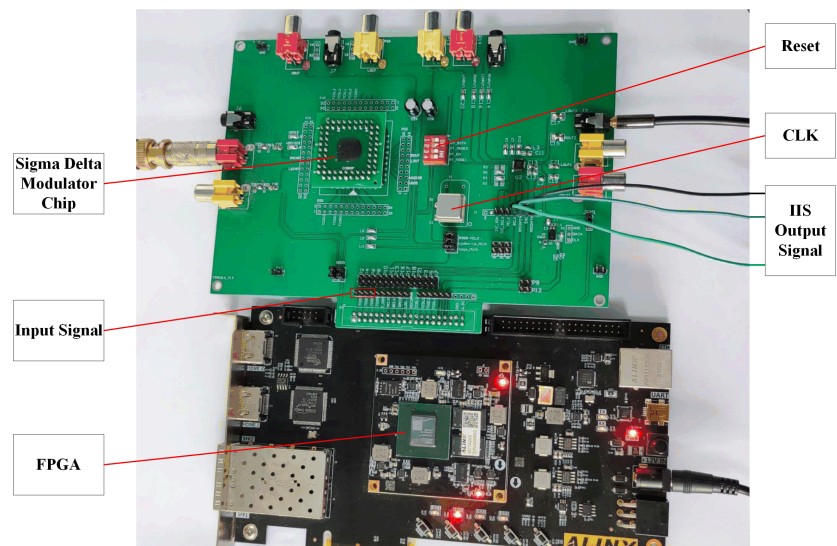

**Figure 8.** Photograph of the FPGA verification platform.

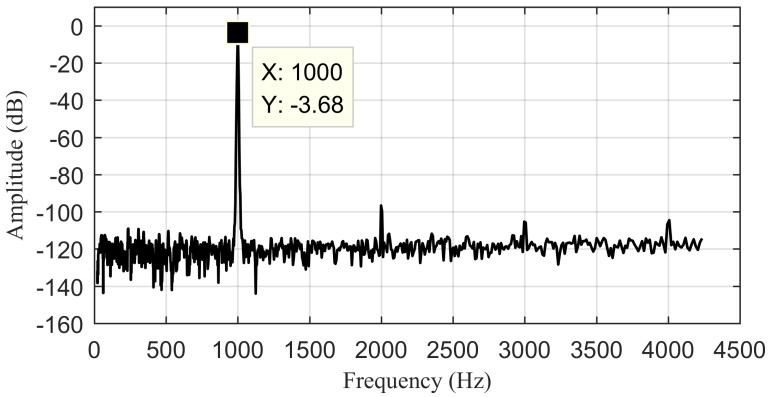

**Figure 9.** Spectrum of the digital decimation filter output signal.

**Table 1.** Performance comparison with similar works.

| Ref. | Chip Model | Architecture | ENOB | Sampling Rate | SNR | Power |
|------|-----------|--------------|------|---------------|-----|-------|
| 3 | AD1871 | Sigma-Delta | 24 | 96 KHz | 105 dB | 368.5 mW |
| 4 | AD7703 | Sigma-Delta | 20 | 4 KHz | 90 dB | 37 mW |
| 5 | AD1556 | Sigma-Delta | 24 | 16 KHz | 120 dB | 20 mW |
| 6 | ADS131M06 | Sigma-Delta | 24 | 32 KHz | 102 dB | 16.7 mW |
| 7 | PCM4220 | Sigma-Delta | 24 | 216 KHz | 123 dB | 340 mW |
| 8 | PCM1804 | Sigma-Delta | 24 | 192 KHz | 111 dB | 225 mW |
| This work | | Sigma-Delta | 20 | 48 KHz | 113 dB | 11.2 mW |

## 5. Layout of the Digital Decimation Filter

The Sigma-Delta ADC contains two parts: Sigma-Delta modulator and digital decimation filter, which are designed independently and implemented on SMIC 0.18 μm COMS process. The Sigma-Delta modulator layout is shown in Figure 10, which is mainly composed of the clock module, power module, third-order quantizer module, reset module, comparator module, and coding module. Finally, the area of the layout is about 714.8 μm × 337.4 μm.

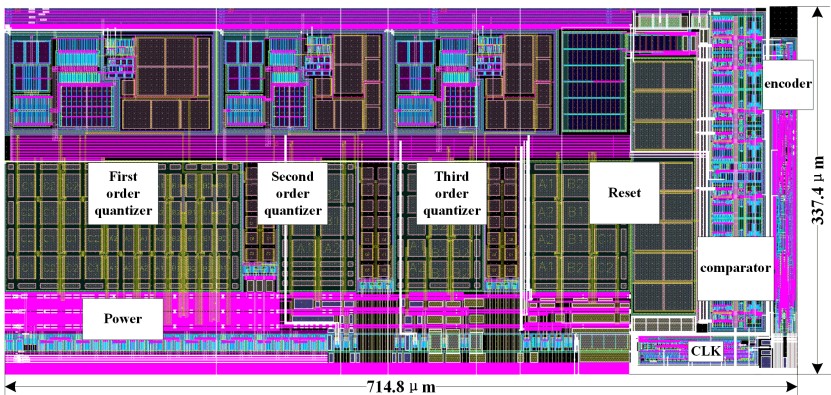

**Figure 10.** Layout of the Sigma-Delta modulator.

After the RTL module of the digital decimation filter is logically synthesized to generate a gate level netlist, and then the Encounter software of Cadence Company is used for the automatic placement and routing (Auto Place&Route) to obtain the layout of the digital decimation filter as shown in Figure 11. The digital decimation filter is made up of a CIC decimation filter, FIR compensation filter, and the half-band filter as marked in Figure 11.

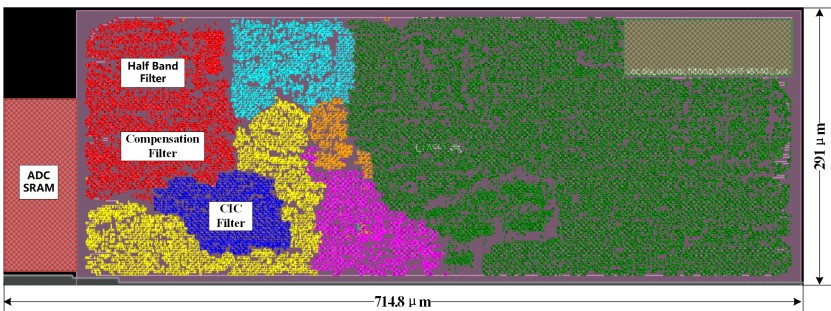

**Figure 11.** Layout of the digital decimation filter.

## 6. Conclusions

This paper presents and verifies a digital decimation filter with a three-stage structure. The first stage is composed of a CIC decimation filter, the second stage is made up of an FIR compensation filter, and the third stage is a half-band filter. In order to meet the design requirements that the SNR is not less than 120 dB and the ENOB is not less than 20 bits, using FDAtool simulates and optimizes its structure. The three-level filter model is successively described by the RTL behavior level and simulated by a VCS+Verdi platform, and the output signal is FFT transformed in the system level simulation to obtain the performance of the digital decimation filter, with the SNR of 121 dB and the ENOB of 19.8 bits. The sine wave signal is restored from the output signal to prove the function and performance correctness of the design. In addition to the behavior level modeling and simulation of digital filters, prototype verification is also carried out on FPGA. The output signal of the Sigma-Delta modulator chip is filtered and decimated by the three-stage decimation filter. The measured SNR and ENOB of ADC are 113 dB and 18.5 bits, respectively. The reasons for the performance degradation are analyzed. Finally, the Sigma-Delta ADC is tapped-out on an SMIC 0.18 μm CMOS process with the layout area of 714.8 μm × 628.4 μm and the power consumption based on backend simulation of about 11.2 mW.

**Author Contributions:** Conceptualization, R.W. and W.Z.; Formal analysis, F.Y.; Funding acquisition, W.Z.; investigation, F.Y.; Project administration, R.W. and W.Z.; Resources, Y.L. and J.W.; Software, C.T.; Supervision, R.W. and W.Z.; Visualization, Y.L.; writing manuscript, W.Z. and C.T.; reviewing and editing, R.W., S.T. and W.D. All authors have read and agreed to the published version of the manuscript.

**Funding:** This work is supported by Key Laboratory of Quark and Lepton Physics (MOE), Central China Normal University under Grant No. QLPL2022P01.

**Data Availability Statement:** Not applicable.

**Conflicts of Interest:** The authors declare no conflict of interest.

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
