# Peer review of "Design and Implementation of Sigma-Delta ADC Filter"

_electronics, doi:10.3390/electronics11244229_

Round 1

Reviewer 1 Report

My comments:

1- all abbreviation over the manuscript such as CDS,VCS,CIS,ADC, etc. must be defined.

2- In line 30, "the power, area consumption ..." is not correct.

3- In line 35, "when choose" does not sound.

4- comparison table must be added.

5- introduction section is weak. similar works must be described.

Author Response

  1. Modified, Detailed please see the blue characters in manuscript.
  2. Modified, "Power consumption, area and conversion accuracy are very important technical specifications of the Sigma-Delta ADC"
  3. Modified, "When the structure of the digital decimation filter is determined, it is necessary to consider the effect of stopband suppression and the requirement of the design specifications"
  4. Added comparison table with similar works in section 6.
  5. Modified. Detailed please see in the introduction section.

Reviewer 2 Report

The authors present the design and implementation on FPGA of a sigma-Delta ADC.

The abstract is misleading as it seems that the authors did fabricated the device whereas they made a tapeout of the device.

The performances obtained using the fpga is already interesting perse.

The presentation is somewhat strange since the authors set a series of requirements without really stating why they need to be met then perform a series of optimization to reach those requirement and finally observe that the requirements were not reached in the end due to physical and experimental limitations. The latter limitations are totally understandable and are what makes it so hard to get those requirements but the narration of the manuscript is quite strange.

Could the authors modulate their manuscript and reasonably explain what they tried to do and were they hit some limitations. In this way, the reader won't be misled by the introduction and the abstract.

It would be interesting if the authors could comment on wether or not a more advanced node could improve the performances like the sampling rate at a constant SNR and ENOb?

In the paper "Chen, K., Chen, M., Cheng, L. et al. A 124 dB dynamic range sigma-delta modulator applied to non-invasive EEG acquisition using chopper-modulated input-scaling-down technique. Sci. China Inf. Sci. 65, 140402 (2022). https://doi.org/10.1007/s11432-021-3401-6" the performance are slightly better but with lower sampling rate. Could the author indicate how their design compare to this somewhat equivalent architecture and on how their approach is innovative.

Author Response

  1. We fabricated the Sigma-Delta ADC on SMIC 0.18 μm CMOS process and tapped-out it. Our main works focus on digital decimation filter. FPGA is used to verify the RTL code of digital decimation code.
  2. The FPGA is only used to verify the function of RTL code of digital decimation filter.
  3. The requirements of SNR not less than 120 dB and ENOB not less than 20 bits are the design targets. We used FPGA to verify the function of the digital decimation RTL code and evaluate the performances of the digital decimation filter. The Sigma-Delta ADC achieves 113 dB SNR and 18.5 bits ENOB on account of noise, interference, and low power supply rejection of the Sigma-Delta modulation chip. The more tests for the fabricated chips will be performed in the future to verify that the Sigma-Delta ADC complies with the design specifications.
  4. For the same circuit structure, more advanced node implies that the chip utilizes lower power supply voltage and occupies lower power consumption. Lower. The lower power supply voltage and lower power consumption means that the chip can achieve higher SNR as well as ENOB.
  5. This paper presents a wide dynamic range sigma-delta modulator with chopper-modulated input-scale-down (CM-ISD)technique that can conquer large input offset while extending dynamic range and obtain higher SNR. We think that the CM-ISD technique is the innovation point of this paper.

Round 2

Reviewer 1 Report

My comments are answered. I suggest acceptance of this manuscript.